# “We Should Be Working Together, and It Felt like They Disrupted That”: Pregnant Women and Partners’ Experiences of Maternity Care in the First UK COVID-19 Pandemic Lockdown

**DOI:** 10.3390/ijerph20043382

**Published:** 2023-02-15

**Authors:** Alice Keely, Mari Greenfield, Zoe Darwin

**Affiliations:** 1School of Human and Health Sciences, University of Huddersfield, Huddersfield HD1 3DH, UK; 2Department of Women & Children’s Health, King’s College London, London WC2R 2LS, UK

**Keywords:** pregnancy, parenting, partners, perinatal mental health, COVID-19, maternity care

## Abstract

Pregnant women were identified as being at elevated risk from COVID-19 early in the pandemic. Certain restrictions were placed upon birth partners accompanying their pregnant partner to in-person maternity consultations and for in-patient maternity care. In the absence of a central directive in England, the nature of restrictions varied across maternity services. Eleven participants (seven pregnant women and four partners), who were expectant parents during the first UK COVID-19 pandemic lockdown, took part in serial interviews in pregnancy and the postnatal period. Data were subject to a reflexive thematic analysis. Four main themes were identified, with sub-themes: uncertainty and anxiety (uncertainty and anxiety about COVID-19, uncertainty and anxiety about maternity services); disruption of partnering and parenting role; complexity around entering hospital spaces (hospitals offering protection while posing threat, individual health professionals in inflexible systems); and attempting to feel in control. Separating couples may result in disruption to their anticipated roles and significant distress to both partners, with potential impacts for mental health and future family relationships. Trauma-informed perspectives are relevant for understanding parents’ experiences of maternity care in the pandemic and identifying ways to improve care to promote and protect the mental health of all parents.

## 1. Introduction

The COVID-19 pandemic brought global threat to public health. National responses varied between and within countries; however, most governments introduced ‘lockdowns’ which included stay-at-home orders and changes to statutory services. In the UK, people were subject to lockdown and social distancing measures and other restrictions, and some were offered specific clinical advice, such as shielding, based upon clinical vulnerability to COVID-19 [1].

NHS Trusts (i.e., health organisations) were not initially given direction from central government regarding measures to control the spread of the virus within maternity services. Thus, while NHS Trusts imposed changes to these services, there were variations between different regions and hospitals. Within maternity services, this included the roll-out of telephone appointments for some aspects of care, restrictions on home visits by midwives and other health professionals, restrictions on birth partner attendance at non-acute appointments, and visiting restrictions during antenatal and postnatal admissions.

In addition to maternal and infant mortality and physical and morbidity risks [2,3,4], research has consistently found that the pandemic has seen increased vulnerability to perinatal mental health difficulties (i.e., during the period from conception to one year after birth). Perinatal mental health difficulties are common and can affect any parent or person pursuing parenthood. Approximately one in five birthing mothers [5] and one in ten non-birthing fathers experience perinatal mental health difficulties [6,7]. Critically, without timely access to effective support, these difficulties can have an impact on all family members’ relationships and mental health, including transgenerational impacts; it has therefore been argued that there is “no health without perinatal mental health” [8].

Rapid reviews examining perinatal mental health in the context of COVID-19 [1,9] have found evidence of heightened psychological distress in the perinatal period, including rates of elevated anxiety and depression symptoms, linked to a lack of information, reduced social interactions, and fears for their own and unborn baby’s health, combined with challenges in accessing services and support.

Most research undertaken concerning perinatal experiences in the context of the COVID-19 pandemic has focused on birthing women’s experiences, although exceptions exist, e.g., [10]. Where any expectant and new parents have been able to participate, for example in surveys, the data have not been linked and have not captured perspectives within couples. To address this gap, the aim of this research was to explore, within a wider lens of perinatal mental health, pregnant women’s and partners’ experiences of maternity care during the third trimester and birth in the first COVID-19 lockdown in England.

## 2. Materials and Methods

We conducted an interpretive qualitative study with pregnant people and their partners in England during the first UK COVID-19 pandemic lockdown in 2020. The key dates of the study’s recruitment and data collection phases are shown, alongside key UK pandemic measure dates, in Appendix A.

### 2.1. Sample and Recruitment

The sample was recruited from a larger sample of new and expectant parents who had completed an online survey about birth experiences early in the initial UK lockdown [11]. To be eligible for the larger survey, participants had to either have given birth since 9 March 2020, be pregnant with an estimated due date of between 9 March 2020 and 3 July 2020, or be the partner of someone who met those criteria. The survey provided a sampling framework of participants who had expressed an interest in participating in a further in-depth interview study, with an additional eligibility criterion of being pregnant/their partner being pregnant and in the third trimester of pregnancy (after 28 weeks’ gestation) at the time of the study’s antenatal interviews (i.e., May–July 2020).

Participants in a prior online survey [11] who had provided permission to be contacted about related research (approximately 500) were informed of the research. It was not possible to only invite those whose due dates were likely to meet the criteria, therefore the email asked individuals to contact this study’s lead researcher via email if they met criteria concerning due date and were interested in taking part in this study. This study’s lead researcher then confirmed eligibility by email, either declining the potential participant or sending an information sheet and providing an opportunity to ask questions, both to them and to their partner, if relevant. All participants signed an electronic consent form and provided demographic details by email. Study materials were tailored for pregnant participants and their partners.

### 2.2. Data Collection

Interviews were conducted May–October 2020 with each lasting between 35 and 90 min. Interviews were semi-structured, following a broad topic guide, with open questions and follow-up prompts indicated. Topic guides were tailored to pregnant and non-pregnant partners and between antenatal and postnatal interviews. Example topics included experiences of lockdown, information provision, pregnancy and birth plans, experiences of maternity care and changes to services, including restrictions concerning partners. For example, sample questions relating to restrictions included:

NHS Trusts are placing restrictions on partners attending for appointments and on being in hospital or birth centre apart from during labour and birth. What’s been your experiences of this? How do you feel about this? Do you understand why these changes have been made? What has this meant for you and your plan for your birth? What have you and your partner discussed?

Sample questions relating to COVID health risks included:

Pregnant women have been identified as at increased risk if they contract the virus. Can you describe your feelings about this? What advice/information have you received, and who from? Do you understand government guidance about pregnant women and how this relates to your partner? What information have you had from your local midwife/NHS Trust? Was this useful or not useful? Have you and your partner had the chance to ask questions?

We aimed to complete an interview with each participant in the third trimester of each index pregnancy (after 28 weeks), then a further interview during the six weeks following birth. The serial interview approach was chosen to capture real-time experiences and feelings during pregnancy [12].

The period of change was twofold for participants, becoming parents during this time, and in a fast-changing world during a pandemic. The research approach was intended to capture anticipated change contemporaneously rather than retrospectively [13]. For context, initial pregnancy interviews took place during May–July 2020, when initial highly restrictive lockdown measures were in place. Postnatal interviews took place during July–October when some easing of the first lockdown measures had occurred (see Appendix A). Hospital restrictions on partners and visitors remained in place throughout this period, but varied somewhat between different NHS Trusts.

All interviews were conducted remotely via Skype (by lead researcher), audio-recorded, and transcribed verbatim. Although interviews were conducted individually, with participants requested to be alone in a quiet room during the interview, partners and others were sometimes present (although not actively contributing), partly reflecting the context of lockdown.

### 2.3. Data Analysis

A reflexive thematic approach was used [14]. The lead researcher completed the initial coding, analysing each transcript line by line and using a hand coding system. Initial codes were refined and grouped into early themes which were then discussed within the research group, followed by further coding by the lead researcher. Preliminary themes were then discussed within the research team, and were refined and finalised.

## 3. Results

Ten pregnant women made contact. All seven who were eligible went on to participate. No birth partners from the original survey made contact; however, all interview participants were encouraged to invite their partners to participate. Four male partners took part; the remaining three declined to be interviewed and did not provide reasons. Of the 11 participants, nine completed two interviews, giving a total of 20 interviews.

One woman had already given birth at the time of recruitment, so only took part postnatally and one partner was unavailable for a postnatal interview. All participants were interviewed individually; however, due to the nature of videocalls and in a period of people living in close confinement, partners were sometimes present during interviews in which they were not actively participating.

Recruitment to the study was UK-wide, however all participants were currently resident in England. All were married or co-habiting, and white British, with the exception of one white European pregnant woman. Four participants (two couples) were first-time parents. The number of previous children was asked of the women and it is not known whether any male participants had children from previous relationships. Further information about participants is provided in Table 1.

### 3.1. Qualitative Results

Four themes were identified, which provided insights into the impact of the COVID-19 pandemic and changes to maternity services on the experiences and well-being of pregnant participants and their partners. These were: uncertainty and anxiety; disruption of partnering and parenting roles; complexity around entering hospital spaces; and attempting to feel in control.

### 3.2. Uncertainty and Anxiety

All participants spoke about the uncertainty that they experienced when faced with limited or rapidly changing information, including from services, from practitioners and in the wider media. Uncertainty was accompanied by feelings of vulnerability, powerlessness and helplessness. Both pregnant participants and their partners described seeking information from a variety of sources (e.g., hospital webpages, central NHS information, social media). Their uncertainty concerned two main area areas: the spread and risks of COVID-19, and the consequent changes to maternity services.

#### 3.2.1. Uncertainty about COVID-19

All the participants said that they had needed information about COVID-19, either in terms of potential risks to themselves, their partner or their baby, or in terms of the likely impact that COVID-19 would have on their perinatal care and birth choices. Most participants said they had deliberately sought out information about COVID-19 at some point before the first interview. The initial interviews took place at a time when information about how COVID-19 might affect pregnancies was scarce, and some participants had difficulty finding trustworthy information:

‘Yeah, most of our information, most of my understanding of COVID has come from the media and, like, they’re not usually the most reliable of places.’(John, Family 3: antenatal interview)

Others employed specific strategies to identify information they felt might be more reliable. A commonly employed strategy was to limit the sources that information was accessed from, selecting only a few trusted sources:

‘I’ve been trying really careful to just following like the same couple of sources, which is basically, yeah, like the Royal College and like the Midwifery, like official sources and the Gynaecologists and the Obstetricians sources, because otherwise I just get really overwhelmed with like social media stuff and I read these terrible stories.’(Beatrice, Family 2: antenatal interview)

The purpose and results of this information-seeking was different for different participants. For some, information-seeking was a strategy they employed to try to manage their anxiety. Other participants described information-seeking for practical purposes, but then found the information available online overwhelming in both volume and content. Some of these participants found reading information about COVID-19 increased their anxiety:

‘I was clicking on all those [NHS webpage] links madly and trying to kind of read what the guidance was and yeah, and I tried, I tried at that point, because obviously I suddenly became massively anxious.’(Hazel, Family 1: antenatal interview)

Amongst those participants who found that reading information increased their anxiety, a strategy employed was sharing the information-seeking task within the couple. Donna felt increasingly anxious, so did not seek news or other information herself, instead she asked her husband to read information and filter it for her. Echoing the language about protection from the virus, she described this as ‘shielding’ herself:

‘I shielded myself quite a bit from the news and let [husband] be my shelter, basically because it was not doing me any good to hear all that negativity. I didn’t really want that.’(Donna, Family 5: postnatal interview)

#### 3.2.2. Uncertainty about Maternity Services

As the lockdown began, individual NHS Trusts made swift and varied changes to services, and many made multiple changes within a short period of time. Several participants experienced confusion and frustration in their attempts to find information about services and restrictions. This heightened feelings of vulnerability for several. Beatrice said:

‘But they said because of COVID that they… they can’t guarantee that we’d be able to go on the midwife led unit, because they would be using the rooms for COVID patients if there were any. Which again is frustrating.’(Beatrice, Family 2: antenatal interview)

The changes introduced by NHS Trusts included the cancellation of some elective caesarean births, the closure of midwife-led units, removal of some analgesic options (i.e., waterbirths and entonox), and the suspension of some homebirth services. As five couples in this study had originally planned homebirths, the availability of homebirth services was of particular concern:

‘I feel like it was just like a lot of things were quite up in the air and there was like a lot of anxiety around all of this and I just felt quite stressed that it had like suddenly just been cancelled and if anything it made me want a home birth more, like obviously with like COVID and hospitals and stuff.’(Colette, Family 7: postnatal interview)

Another participant described her anxiety that her local hospital could not reassure her beyond a statement that, on the day she went into labour, the homebirth service might—or might not—be available:

‘I guess that’s the thing that’s in my mind which is that I could call on the day and they could say, “no, you have to come in”. Which obviously could always happen, but it’s just more likely…’(Beatrice, Family 2: antenatal interview)

The uncertainty caused by frequent changes to service provisioin and differing responses to the pandemic between neighbouring NHS Trusts frustrated many participants. This was compounded in some cases by a lack of clear communication about both service availability and the rationale for the changes. These frustrations were not only experienced by the pregnant women who were interviewed. Partners also described feeling frustrated at the lack of available information, and the effort required of them to secure information:

‘Up until March [2020] I’d heard the midwife saying at our appointments that I went to, saying a home birth is absolutely fine in our area, and then they got taken off the table because of COVID. And then things start calming down a little bit towards the end of May, June. I thought, right, and [partner] tried to get hold of the Head Midwife, but there was no contact details for her anywhere so [partner] ended up going through the complaint service of the hospital and then from that the community midwife phoned and explained why home births weren’t on.’(Tim, Family 1: antenatal interview)

Conversely, a small number of participants experienced good communication about service changes and found this beneficial. One woman, who planned to have her baby in hospital, discussed how her midwives had communicated with all expectant parents on a planned and regular basis via social media, providing her with the opportunity to access information and ask questions. She said:

‘So, they have been, they have been good at, sort of, supplying information... every Friday. The midwifery team do a Facebook Iive chat, which has been really, really helpful actually. That’s been really good.’(Joanna, Family 4: antenatal interview)

Not only could she access this information, but so could her husband. As the local restrictions regarding antenatal appointments meant that he could not attend appointments and ask questions there, this form of communication was helpful in ensuring that he both felt involved in the perinatal care, and had the information he needed about how they changes impacted his role. From watching these live chats, Joanna’s husband explained what he understood the restrictions to be:

‘When she is having the baby, I’ll pretty much drop her off and then they’ll give me a call when she’s in (..) established labour if you like. Then I can come into the hospital for the birth and then maybe spend a little bit of time with them after, and then I have to leave pretty quickly.’

When asked how he felt about this, he said:

‘I completely understand it, I do. I mean I would love to be there for the whole thing, I mean, it was an amazing experience with [1st baby], but I’m not, kind of, cross about it. I’s just the situation’(Graham, Family 4: antenatal interview)

The reassurance this couple received from knowing both that there was a regular pattern of communication about service changes and understanding the restrictions which were in place and the rationale for these highlights how important access to reliable information about service changes was to expectant parents. Joanna’s description of communication from the local maternity service as ‘helpful’ is in sharp contrast to the confusion, frustration and lack of trust expressed by other participants. This may in part be related to planned place of birth, however, as uncertainty around homebirth provision was highlighted as a particularly difficult issue.

### 3.3. Disruption of Partnering and Parenting Roles

Many of the participants described the service changes as reducing their choices and compromising their decision-making. These changes were viewed as individual NHS Trusts imposing rules and restrictions, even when participants also expressed an understanding of the reasons for the changes. Participants expressed feeling powerless in the face of these large institutions, who they could not hold accountable:

‘We’re not even able to make decisions ourselves. Everything is sort of being dictated to us and the lack of communication, as well for [partner] and, you know, she has these questions and then they’re not getting answered.’(Seth, Family 2: antenatal interview)

As choices were restricted, individuals’ and couples’ preferred roles—both as parents and as partners—were disrupted. A major cause of disruption was the physical exclusion of partners from most appointments, which they would otherwise have attended. Women were required to attend all maternity appointments alone, with the exception of when they were giving birth. Restrictions applied to routine appointments, emergency appointments for particular concerns, and appointments for induction of labour or early labour. After the birth of their baby, Seth recalled his feelings on being excluded from antenatal care:

‘I felt like I could have really helped a lot more if I was in the appointment, I could have offered my insights and also my perspective on like her sleeping pattern and the way the baby was kicking, just everything that I felt like I actually could have had a positive imprint on them appointments, if I was there too.’(Seth, Family 2: postnatal interview)

This physical exclusion meant partners were unable to provide support, or to receive information and support from healthcare professionals (HCPs). One partner described how important being able to fulfill this role was to him:

‘Generally, my main desire in life is to support [partner] and to make sure she’s okay and to be that supportive partner… So yeah, it was about just being supportive.’(Tim, Family 1: postnatal interview)

Despite expressing an understanding of the restrictions, being unable to carry out this role caused him distress. It also made him feel excluded as a parent from discussions that led to decisions that affected his baby:

‘But then, you know, I did begin to become frustrated around actually these are big decisions and I’m not there to be able to make them… We should be working together, and it felt like they disrupted that quite a lot.’(Tim, Family 1: postnatal interview)

Emotions were heightened further for this couple when pre-eclampsia (a serious pregnancy complication) meant Hazel was admitted to hospital.

Hazel described that she was given this worrying diagnosis and then expected to communicate it to her husband herself. Not only were HCPs not available to explain the situation to her partner, but she had to communicate the diagnosis over the phone, as her partner was not allowed into the hospital, and as she had been admitted, she felt she was not allowed out. She felt this was inappropriate and unacceptable:

‘Like, I shouldn’t have had to sit there and tell [husband] over video that I had pre-eclampsia, like that. He should have been there while I was told that, you know, he should have been part of that conversation.’(Hazel, Family 1: postnatal interview)

After her admission, Tim was still not allowed into the hospital to support Hazel. He described spending long hours in the hospital car park, not wishing to leave the hospital site, but unable to enter the building and see his pregnant wife. He said:

‘There’s nowhere to wait because the waiting rooms and the café and everything is closed. You can’t... so you have to sit in the car or wander round the streets. So that’s what you found outside [the hospital], there was just lots of men walking around the car park.’(Tim, Family 1: postnatal interview)

Later, when Hazel was asked to make decisions about her care, the couple tried to include Tim in this decision-making via mobile phone. He described how unsatisfactory this experience was for him and how little information was given to him, clearly voicing how much the experience disrupted his expectations of the role he would play as both a partner and a parent:

‘I was aware of this guy stood at the end of the bed kind of saying ‘you need to sign this because you’ve agreed to..’ whatever it was, but I didn’t, I still don’t really know what that was about and it just felt as though, you know, this is a decision to do with my wife and my child and I can’t really hear what’s going on. I don’t know what the form is, I can’t see what it says on it.’(Tim, Family 1: postnatal interview)

It is notable that perspectives about the disruption to partner and parent roles varied amongst participants. As they decided where to give birth, another couple sought information about the restrictions at their two local NHS Trusts and the risks of contracting COVID-19 while in hospital. Describing their discussions, Emily said about her husband:

‘[H]e’s got quite a lot of trust and faith [that] in either hospital the risks will be managed by the people that work there.’(Emily, Family 3: antenatal interview)

John confirmed this representation of his approach. During the first interview he saw his role as being to protect his partner and unborn child by taking advice from those with more expertise, and following their advice:

‘I’m not a medical professional, I’m not an epidemiologist or virologist, I don’t kind of question too much the guidance. I just think it’s best practice just to follow it. Let’s just all be careful and follow the rules.’(John, Family 3: antenatal interview)

These different expectations about parent and partner role meant John mostly viewed the changes as enabling him to fulfill his role, rather than as impositions from the NHS Trust which disrupted his role. Despite this, he did indicate some feelings of conflict about the consequences of the changes. Contemplating the possibility that he might not be able to be present at the planned induction of labour in a few days’ time he said:

‘Induction is hard for the woman. It’s really hard and to do that on your own is, you know, it’s just, it’s not ideal is it? It’s much better if someone is there to hold your hand… The most important thing is that, yeah, you get a baby out safely and the mother’s safe and that we don’t unnecessarily spread anything around.’

He was also aware that his feelings when thinking about not being able to be with his partner might be different to the reality of actually not being present at the time. Later he added,

‘I mean probably right now, because we’re not in the thick of it, I’m able to see that bigger picture and be all philosophical about it. You know, at crunch time, I might feel a bit differently, who knows. You know, if it goes badly wrong, I don’t know, you just don’t know how you’ll feel.’(John, Family 3: antenatal interview)

For most participants, the restrictions imposed on partners disrupted the roles they had expected to play in relation to each other and to their unborn baby. For those who were pregnant, their role changed to that of being a conduit of information to their partner, whilst their partners were unable to offer information or be involved in decision-making that affected their baby and were not able to offer the support to their pregnant partner that they wished to, thus contributing to both feelings of helplessness for partners and to women feeling unsupported.

### 3.4. Complexity around Entering Hospital Spaces

Participants described their complex feelings about what going to hospital might mean for them. Traditionally viewed by most expectant parents as the safest place to give birth, these institutions now simultaneously posed a threat to their safety. Within this context, however, individual interactions with caring HCPs were recounted.

#### 3.4.1. Hospitals Offering Protection While Posing Threat

Several participants referred to the threat they perceived in attending a hospital, either when giving birth, or antenatally for routine appointments or because there was a concern about the pregnancy. One participant described her difficult decision-making process when she felt her baby’s movements had changed in nature during her third trimester—something which pregnant women are advised to seek clinical attention for:

‘I think pre-COVID stuff, I probably would have gone, but it suddenly felt so much more complicated and risky to go, because I was thinking, well, if I go down to the hospital, first of all my husband can’t come with me because there’s no-one to look after the children. So, I’d have to go on my own, which would mean me getting in a taxi and there’s a degree of risk in getting in a taxi and then going into the hospital and being on my own in the hospital. There’s a degree of risk of me picking something up and bringing it back.’(Joanna, Family 4: antenatal interview)

Joanna perceived that seeking help from the hospital offered her a mix of protection and simultaneously threatened her safety. Balancing these dissonant views led her ultimately to a decision not to attend the hospital on this occasion. Other participants described similar difficulties in deciding whether to attend hospital. Those who did choose to attend hospital described facing similar mixed feelings of threat and protection, for example feeling vulnerable attending appointments alone when they could not maintain social distancing:

‘There were security guards at the gate of the car park. It was really quiet and they came really close, they asked me to wind down my window and they came really close and I just felt really like, this feels really strange, Like, I appreciate you need to check something, but you’re getting really close to me and presumably you’ll see everyone coming into this hospital. So, I was a bit like okay, alright, back off!’(Donna, Family 5: antenatal interview)

Others expressed an awareness that hospitals represented the setting for both the births of babies and of the COVID-related deaths of acutely ill patients—a fact which made many participants uncomfortable as they feared for the life of their baby and/or partner. One partner said:

‘I don’t want [partner] having our baby in a hospital where there might be a COVID ward just down the corridor.’(Tim, Family 1: antenatal interview)

Hospitals are traditionally seen as providing care and offering safety but were perceived by the participants in this study as also posing risk and a potential threat to the life of their unborn child. This altered perception left many participants feeling that there was no safe place for their baby to be born.

#### 3.4.2. Caring Individuals in Inflexible Organisations

Whilst poor communication, confusing restrictions, and challenging experiences were repeatedly noted by participants related to receiving care in hospital, positive individual care interactions also occurred. When this was provided in circumstances participants observed as difficult, it was appreciated:

‘[T]he midwives were brilliant and really helped her through it, and then they were under a lot of pressure. They were really busy.’(John, Family 3: postnatal interview)

Participants did not view individual HCPs and the institutions they worked for as interchangeable. Rather, they saw that the institutions were imposing restrictions on both themselves and the HCPs. Many perceived HCPs to be struggling against both the imposed restrictions and additional the additional risks associated with COVID-19 in a similar way to themselves:

‘The midwives, you know, they’re wonderful, they are generally kind and empathic and they understand, they listen. It’s the system, it’s the structures, it’s the hierarchy, it’s the power imbalance that is the difficulty.’(Tim, Family 1: postnatal interview)

With the withdrawal of home visits in some NHS Trusts, and the partial stoppage of health visiting services, hospitals were, for many participants, the only places that pregnant people could be in face-to-face contact with HCPs. This added complexity to participants’ views of hospitals as places of both safety and threat, as they were the (risky) location where caring HCPs could be accessed.

### 3.5. Attempting to Feel in Control

Participants described multiple ways in which they attempted to gain control over their perinatal experiences in the context of the disruption imposed by both the pandemic and the maternity service changes, which included seeking information (as noted earlier).

One participant hired an independent midwife as her uncertainty deepened regarding the availability of NHS care during a homebirth. She later described her positive feelings about having taken this decision, which had caused financial strain to her and her partner:

‘I’m just so pleased, you know, that I made that decision about having the private midwife in the end and everything, and being able to do it at home and you know, because I sort of feel a birth is such a deep-rooted experience and you know, you only have really that, your memory for a lifetime, so it is just so important that it should be the most positive experience.’(Inge, Family 6: postnatal interview)

Another participant described how she decided to accept an induction of labour. This would not have otherwise been her preference, but in the context of COVID-19, the option of an induction offered some control over when her baby would be born, affording certainty in her childcare arrangements and facilitating her husband’s presence at her labour and the birth of their baby:

‘I think for me it was really mixed, and the reason I decided to go ahead with it was because of what the induction offered in terms of certainty in a very uncertain world. That I had a date when I knew I needed my children looked after and a date when there was a ticking clock in terms of when my baby would arrive.’(Emily, Family 3: postnatal interview)

However, this decision resulted in a subsequent loss of control in other areas. Before the induction, Emily attended her final antenatal appointment alone, and underwent a membrane sweep, an internal procedure intended to expedite labour, or to contribute to the ease of the planned induction process. She described her experience of this at her postnatal interview:

‘I had said ‘Stop’ and I probably would have been more forceful, you know, when she said ‘No, I’m just going to carry on’, I would have said ‘No, you’re going to stop’, had it not been for the fact that I just wanted this induction to happen, given that the unit was full, there’s a pandemic, everyone is wearing masks, everyone’s a bit stressed, you know. Those kinds of things that impacted what I decided… when actually my instinct was ‘I’d like you to take your hand out of my vagina now, please.’(Emily, Family 3: postnatal interview)

Emily’s decision to not insist again on the internal procedure being stopped highlighted her complex and vulnerable position. She described seeking certainty about her labour and birth, even if it wasn’t what she would have wanted for her birth experience. She described the fraught circumstances she perceived within the hospital setting as influencing her choice. Finally, she sought to increase the likelihood of support with childcare from family in difficult lockdown circumstances, by choosing induction at a specific time which, crucially, meant she would have support from her partner being there.

Emily later described having a long and difficult labour, reflecting:

‘Had I been more confident about myself and my body… and more confident with the idea of uncertainty… maybe I would have trusted to have not accepted that induction, and that things might have been a bit easier.’(Emily, Family 3: postnatal interview)

Emily’s experiences are an example of participants’ attempts to regain control of their experiences sometimes involved compromises that, due to the uncertain nature of labour and birth, contributed to further losses of control.

## 4. Discussion

This study adds to the growing literature on perinatal experiences during COVID-19, which show a significant rise in perinatal mental health difficulties experienced by those who gave birth during the first UK lockdown [15,16]. Participants within our study experienced feelings of powerlessness and helplessness in the context of the pandemic, disruption to their roles as partners and expectant parents, and dissonance in their perceptions of hospitals. They employed a variety of strategies to cope with these experiences. These findings cohere with wider literature on the pandemic, including expectant parents’ need for reliable information in the face of ‘chaotic’ messaging [17], uncertainty about maternity care restrictions and disruptions to birth plans [18]. In a situation of such uncertainty, including at governmental and health service levels, the increased psychological distress is unsurprising.

This study extends the wider literature in several ways. First, the study shows the ways in which restrictions disrupted individuals and couples in their roles, both as partners and as parents. There were several examples where this was largely negative, for example, hearing from the perspectives of both parents about the distress and frustration experienced in being unable to support a partner who was receiving a health diagnosis or undergoing a procedure. However, it was notable too that some partners took on new ways of providing support, for example, in sharing responsibility for seeking and making sense of information about COVID-19 and changes to services.

Second, this study adds to the emerging evidence about the impact of the pandemic on birth choices. Feeling in control of decisions is established within the literature as a protective factor against birth trauma and for perinatal mental health. In line with other studies [11,17], we can see that following the withdrawal of homebirth services and closure of birth centres, a range of different strategies were employed by participants to regain a sense of control over birth choices, including hiring an independent midwife, considering freebirth, and choosing to have an induction of labour. In this current study, we can see the complex interplay between multiple new restrictions and existing procedures, for example partners admitted only once labour is established alongside vaginal examinations as the standard labour assessment method resulting in participants feeling they had to have a vaginal examination to secure their partners’ presence. Pressure to accept unwanted interventions—a phenomenon sometimes dubbed ‘informal coercion’ [19] or ‘informed coercion’ [20]—is common within maternity services, and we suggest that restrictions on partner presence may have led to more instances of informed coercion. Such practices further compromised some participants’ attempts to regain control of their choices during birth, for example allowing a membrane sweep for which consent has been withdrawn to continue from fear a wanted induction of labour might be refused. There is a sense that as different aspects of choice and control were lost couples tried to make choices to navigate to what they perceived to be the least bad path for birth, only to experience further obstacles, resulting in women unwillingly making decisions alone whilst their partners are excluded from the decision-making.

This study resonates with Menzel’s work [21], which used thematic social media analysis to explore the emotional impacts of visitor restrictions for fathers. Even without such restrictions, it is common for fathers to feel excluded from maternity spaces, which can be accompanied by uncertainty, fear and frustration [22]. Both the current study and Menzel’s work highlight that institutional responses to infection control can undermine partners being seen as parents (here, fathers); described by Menzel as instead being repositioned as “spectators, rather than active participants in pregnancy/parenthood” (p. 85). The potential for visitor restrictions to compromise social recognition as parents has been reported elsewhere [23]. Such approaches appear inconsistent with increasing policy emphasis on being family-centred and supporting the transition to parenthood for all parents, as part of promoting mental health and relationships in the critical 1001 days spanning pregnancy and early years [24].

All of these findings are relevant for promoting perinatal mental health, including through implementing trauma-informed care in the perinatal period. As outlined in the NHS England good practice guide, trauma-informed care aims to “promote feelings of psychological safety, choice and control” (p. 13) and needs to involve consideration of all staff, including for example reception staff [25]. This study’s findings show that maternity care experiences, including inaccessible or inconsistent information, and the imposed restrictions, may directly compromise these aims. In addition, examples showed how the pandemic may further widen the staff involved in the expectant parents’ experiences of birth, giving examples of interactions in car parks and with security staff. The study showed the potential for positive interactions between parents and individual staff—even when organisations felt inflexible and threatening. As outlined in the guide, principles of trauma-informed care should also be applied to partners, and the current study indicates the significance both for partners as individuals and in adequately recognising the importance of a birthing person’s autonomy about having their chosen support people with them at any contact with services.

### 4.1. Ethical Approval

The study was conducted in accordance with the Declaration of Helsinki, and the protocol was approved by the University of Huddersfield School of Human and Health Sciences Research Ethics and Integrity Committee, project code 2020/054. All subjects gave their informed consent for inclusion before they participated in the study.

### 4.2. Strengths and Limitations

This was a small sample and may not have been representative of couples’ experiences during the first pandemic lockdown. Whilst an exploratory qualitative study does not aim for generalisability, it is pertinent to reflect on issues of transferability (the extent to which the results can apply elsewhere). While it is notable that no birth partners from the original study made contact, it is perhaps not surprising given that only just over 1% of the original survey participants were birth partners. A key limitation is the sample’s homogeneity concerning ethnicity whereby all participants were white. There is consistent evidence of health inequalities in perinatal morbidity and mortality, and experiences of maternity services relating to ethnicity [26,27,28]. In addition, people of colour have been disproportionately impacted by COVID-19 [29,30]. Community engagement work highlights the need to consider the different ways in which minoritised ethnic groups may be impacted by the COVID-19 pandemic and service responses [31]. Further research is urgently needed with couples from minoritised ethnic backgrounds. Also relevant for transferability is the high proportion of participants who were considering or actively planning a homebirth prior to the lockdown. Similarly, we recognise that recruitment occurred online and that consideration of digital literacy is needed, especially given this study’s findings about the role of information-seeking in attempting to feel in control.

Our study’s focus concerns the first six months of the pandemic, which may be a time of heightened anxiety. However, other pandemic-related research indicates enduring impacts relating to the pandemic [32] and these experiences also have implications for future reproductive choices and experiences. In addition, we acknowledge that while health services, and to some extent supplies, were disrupted in the UK during this time, the nature and extent is not comparable to the experiences of countries within the global south, for example [33]. Study strengths included the serial interview design, which allowed for more contemporaneous data collection, and the active invitation for non-birthing parents to participate. Through inviting the experiences of both birthing and non-birthing parents, we were able to understand the significance of maternity care experiences for both parents, as individuals and as couples.

## 5. Conclusions

Changes to provision in maternity services warrant holistic consideration both of health and of the family unit. When changes to birth choices are inevitable, also separating couples may result in disruption to their anticipated roles and significant distress to both partners, with potential impacts for mental health and future family relationships. Parental mental health is of equal importance as physical health for the parents, existing children, and the newborn baby. The impact of communication about any necessary changes also requires consideration, with greater attention to ensure that messages are clear, communicated widely, and changes are minimised.

An understanding of trauma deepens the understanding of parents’ experiences of maternity care in the pandemic and helps to identify ways to improve such care. Trauma-informed perinatal care is relevant for all staff and all parents, and is consistent with the importance of perinatal mental health for all families.

## Figures and Tables

**Table 1 ijerph-20-03382-t001:** Participant characteristics.

Family	Participant	Age	Employment	Number of Previous Children	Original Planned Birthplace	Preference Birthplace at Antenatal Interview	Birthplace	Notes
1	Hazel	30	Public sector	0	Home	Home	Hospital (medical reason)	Booked for homebirth; pre-eclampsia, induced labour; non-operative vaginal birth
	Tim	34	Public sector					
2	Beatrice	31	Health worker	0	Home	Home	Hospital (homebirth cancelled by NHS)	Booked for homebirth, considered freebirth when uncertain about service provision; non-operative vaginal birth
	Seth	30	Arts worker					
3	Emily	32	Health worker	2	Hospital	Hospital	Hospital	Induced labour; non-operative vaginal birth
	John	33	Business owner					
4	Joanna	33	Public sector	1	Hospital	Hospital	Hospital	Non-operative vaginal birth
	Graham	33	Manual skilled					
5	Donna	35	Retailer	2	Home	Home	Home (NHS)	Non-operative vaginal birth
6	Inge	39	Home maker	1	Home	N/A *	Home (Independent MW)	Booked for homebirth with NHS Trust but transferred to independent midwifery care amidst uncertainty around service provision. Non-operative vaginal birth
7	Collette	30	Self-employed	1	Home	Home	Hospital (homebirth cancelled by NHS)	Non-operative vaginal birth

* Took part in postnatal interview only.

## Data Availability

Transcripts will not be made available because the combination of contextual information given by participants could compromise their anonymity if the transcripts were available in their entirety.

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
