# Peer review of "“We Should Be Working Together, and It Felt like They Disrupted That”: Pregnant Women and Partners’ Experiences of Maternity Care in the First UK COVID-19 Pandemic Lockdown"

_ijerph, 2023, doi:10.3390/ijerph20043382_

Round 1

Reviewer 1 Report

Thank you for the opportunity to read this very insightful manuscript. I have made some general comments and suggestions below.

1. there were some grammatical issues throughout that should be addressed. For example, please review the last sentence of the methods section.

2. Although a qualitative study can suffice with fewer participants, I am wondering whether additional recruitment methods were utilized? If only four positive responses were obtained why not attempt to recruit via physicians?

3. the timeline for interviewing should be discussed and considered in more detail. Specifically, it would be beneficial, given the history effects, to perhaps create a figure or timeline detailing what critical COVID related impacts were being experienced in the UK during those specific months. My inclination is that during months of heavy uptick you would be more likely to receive anxious responses.

4. one aspect of the study not discussed in the limitations is that this sample is entirely White. Given the heightened infant and maternal mortality rates for people of color, poorer healthcare experiences with OBGYN care, and their increased likelihood to experience poorer COVID related outcomes, it seems pertinent to discuss these implications. This again leads to the question of why more participants were not recruited. If for nothing more than to capture a more diverse sample and perspective. Please explain and discuss.

Reviewer 2 Report

Overall, this study investigates an important topic during the COVID-19 pandemic. However, there are several methodological concerns, I raise them here:

1. Research design
Can you clarify how only seven out of five hundred contacted subjects were included in the study might affect external validity? How does your sample compare to the typical expecting parent population? (i.e. age group, demographics, income-level, etc.) Why did some partners refuse to be interviewed (three out of seven, about half were excluded)?

2. Construct measures
Can you provide additional details on the interview protocol? 

3. Analysis and results
The present results section mainly consists of quotes, additional analyses in linking these quotes would be useful, and provide a context for interpretation.

4. Discussion
The existing discussion is quite weak. Implications of the present study should be elaborated. It would be useful to consider and add the latest literature the following studies that cover the topic of COVID-19 and psychosocial stress, particularly its related causes and consequences of COVID-related health outcomes, generalized to a broader scope. I randomly searched a couple key words relevant to your study on Web of Science and think you should consider citing these relevant studies:

a) https://www.webofscience.com/wos/alldb/full-record/WOS:000733186600001

b) https://www.webofscience.com/wos/alldb/full-record/WOS:000653921900001

Author Response

Please see attached word document

Round 2

Reviewer 1 Report

Thank you for your careful and thoughtful review of the manuscript and suggestions made. At this time all of my concerns and suggestions have been addressed.

Reviewer 2 Report

Thanks for addressing prior comments